# Percutaneous Coronary Angioplasty in Patients with Cancer: Clinical Challenges and Management Strategies

**DOI:** 10.3390/jpm12091372

**Published:** 2022-08-25

**Authors:** Gemina Doolub, Mamas A. Mamas

**Affiliations:** 1Keele Cardiovascular Research Group, Keele University, Keele ST5 5BG, UK; 2Cardiology Department, Bristol Heart Institute, Bristol BS2 8HW, UK

**Keywords:** cancer, percutaneous coronary angioplasty, coronary artery disease, outcome

## Abstract

The number of cancer survivors in the United States is projected to increase by 31% by 2030. With advances in early screening, diagnosis and therapeutic strategies, a steadily increasing number of patients are surviving cancer. Coronary artery disease (CAD) is now one of the leading causes of death amongst cancer survivors, with the latter group of patients having a higher risk of CAD compared to the general population. Our review covers a range of specific challenges faced by doctors when considering percutaneous coronary interventions (PCI) in cancer patients; clinical outcomes in cancer patients undergoing PCI, as well as some important technical considerations to be made when making decisions regarding the management strategy in this special population of patients.

## 1. Introduction

The number of cancer survivors in the United States is projected to increase by 31% to 22.2 million by 2030 [1]. With advances in screening, diagnosis, and therapeutic strategy, an increasing number of patients are surviving cancer. In fact, over the next decade, the number of people living 5 or more years following their cancer diagnosis is predicted to grow by 33% to 15 million [2]. Coronary artery disease (CAD) is now one of the leading causes of death amongst cancer survivors, [3] with the latter having a higher risk of CAD compared to the general population. One study found that more than one in ten cancer patients die of heart disease rather than their cancer, and for some cancers, such as breast, prostate, endometrial, and thyroid cancer, 50% will die from cardiovascular disease (CVD) [4].

Certain overlapping risk factors, such as smoking, pathophysiological, and genetic factors, may additionally account for this observation. Additionally, some therapies used for the treatment of cancer can increase the risk of acute myocardial infarction (AMI). For instance, the use of immunotherapies such as rituximab and bevacizumab, has been reported to correlate with a risk of acute myocardial infarction [5,6]. Furthermore, chemotherapy alkylating agents, such as cisplatin, antimetabolites such as capecitabine and fluorouracil, and vinca alkaloids, are known to cause ischemia. [7] Radiation therapy, when administered to the thorax, has the potential of damaging the myocardium, pericardium, and coronary vessels [8]. Other effects include endothelial cell damage as well as radiation-induced accelerated atherosclerosis.

Percutaneous coronary intervention (PCI) is the most common mode of revascularisation utilised in patients with multiple comorbidities [9], including cancer. This comes with a set of specific clinical challenges, as cancer is associated with a greater risk of mortality and morbidity [10], including conflicting bleeding and thrombotic risks in this particular group of patients, which in turn dictates management in terms of the stent technology used and antiplatelet regime and duration chosen, as well as considerations surrounding technical jeopardy. In this clinical review, we examine the relationship between cancer and CAD, the clinical challenges commonly encountered, and the management strategies to be contemplated when treating cancer patients undergoing PCI.

## 2. PCI in Cancer: Specific Challenges

Coronary intervention for CAD in cancer patients presents some unique challenges. Patients with malignancy are often older [11,12], with a greater comorbidity burden [13,14] and more extensive coronary artery disease [11].

Patients with cancer generally tend to have increased risks of bleeding as well as thrombosis, the latter manifesting as arterial thrombotic events, including AMI and stroke [15]. Cancer can trigger pro-inflammatory and prothrombotic states through the induction of cytokines, altered platelet activity, endothelial dysfunction, oxidative stress, and disorders in coagulation and fibrinolysis [16]. Furthermore, bleeding is a common challenge encountered in cancer patients, which can be related to local tumour invasion, tumour angiogenesis, systemic effects of the cancer itself, or oncology therapies [17]. Additionally, bleeding can be exacerbated by chemotherapeutic agents, such as bevacizumab, non-steroidal anti-inflammatory drugs, and anticoagulants used in cancer patients [17]. Active malignancy (defined as a diagnosis within the previous 12 months or ongoing active cancer therapy, including surgery, chemotherapy, or radiotherapy) is considered one of the major ARC-HBR criteria, as defined by the Academic Research Consortium for High Bleeding Risk [18].

Importantly, radiation treatment in cancer patients can lead to accelerated coronary calcification and thus contribute to a burden of complex calcific diseases in this population [19]. Calcific coronary artery disease (CAD) poses important challenges [20] to PCI by hindering lesion crossing [21] and inhibiting optimal stent expansion and apposition [22].Stent under-expansion is itself a key predictor of stent thrombosis (ST) as well as in-stent restenosis (ISR) and is associated with repeat target vessel revascularisation (TVR). A previous meta-analysis looking at the impact of calcific disease on outcomes following PCI highlighted the fact that severe coronary calcification resulted in less complete revascularisation (48% vs. 55.6%, *p* < 0.001), and was associated with higher mortality (10.8% vs. 4.4%, *p* < 0.001) and higher rates of coronary revascularisation (31.8% vs. 22.4%, *p* < 0.001) [23]. For this reason and in an effort to improve outcomes in these patients, an emphasis has been laid in recent years on optimising calcium modification by means of technology such as rotational atherectomy and, more recently, intravascular lithotripsy.

## 3. PCI Outcomes in Cancer Patients

There are limited data on clinical outcomes following PCI in cancer patients, and this is partly because most randomised controlled trials have traditionally excluded patients with active cancer. From the available data, studies have suggested that a diagnosis of cancer is associated with death [11,14,24], acute myocardial infarction (AMI), [25] major bleeding [26], and target lesion revascularisation [27]. A recent meta-analysis looking at outcomes following PCI in cancer examined a total of 33,175 patients undergoing PCI, of whom 3323 had cancer. At one year, the patients in the cancer group had significantly higher mortality (RR 2.22, 1.51–3.26, *p* < 0.001), including cardiovascular mortality (RR 3.42, 1.74–6.74, *p* < 0.001), compared to the non-cancer patients. [28] In a large database study looking at 1.9 million patients undergoing PCI, the 90-day readmission for AMI following PCI was higher in patients with active cancer (9.1% vs. 5.6%, *p* < 0.001). [25] This particular study also looked at bleeding rates and found higher bleeding rates amongst patients with cancer (1.6% vs. 0.6%, *p* < 0.001).

From the BleeMACS registry [13], it has been seen that patients with cancer undergoing PCI are more likely than their non-cancer counterparts to suffer composites of death or reinfarction (15.2% vs. 5.3%, *p* < 0.001), as well as bleeding (6.5% vs. 3%, *p* < 0.001) at one year. Large population-based studies, such as the one by Potts et al. [24], have reported a two-fold increase in in-hospital mortality and complications such as major bleeding in patients with lung cancer. Colon cancer was associated with the greatest risk of major bleeding (OR 3.65, 95% CI 3.07–4.35). An explanation provided by the authors for this finding includes cancer-specific responses to DAPT regimes following PCI [24]. Additionally, and perhaps unsurprisingly, metastatic disease was independently associated with mortality, PCI complications, and bleeding events. Metastatic lung cancer in particular was associated with a four-fold increase in mortality. Poor outcomes following PCI do not seem to be limited to solid organ malignancy but also to haematological malignancies. Indeed, a large NIS-based study [29] looking at leukaemia patients (of which chronic lymphocytic leukaemia (CLL) represented 75% of the cohort) showed that patients with a current diagnosis of leukaemia were at significantly increased odds of in-hospital mortality (OR 1.41, 95% 1.11–1.79), as well as bleeding (OR 1.87, CI 1.56–2.09), with patients with acute myeloid leukaemia (AML) having a five-fold risk of in-hospital mortality [29]. In another database study [30] looking at the outcomes of cancer patients in STEMI, it was reported that only 30.8%, 20.2%, and 17.3% of breast, lung, and colon cancer patients received PCI. Amongst patients without any of these cancers, the use of PCI was nearly 50%. In-hospital mortality was highest in patients with lung cancer (57.1%) vs. 25.7% in non-cancer patients.

Cancer patients undergoing PCI are also at greater risk of bleeding, thrombosis, and repeat revascularisation. Another large population study [31] analysed trends in outcomes and gastrointestinal bleeding (GIB) associated with PCI, and found that independent predictors of GIB included rectal/anal cancer (OR 4.64, 95% CI 3.20–6.73, *p* < 0.0001), stomach cancer (OR 2.74; 95% CI 1.62–4.66, *p* = 0.0002), oesophageal cancer (OR 1.99; 95% CI 1.08–3.69, *p* = 0.0288), and colon cancer (OR 1.69; 95% CI 1.43–2.02, *p* < 0.0001). [31] From the Dutch Stent Thrombosis Registry [32], it was suggested that the presence of malignancy was independently associated with stent thrombosis (OR 13.1; 95% CI 1.99–85.03, *p* = 0.0074). The Kumamoto University Malignancy and Atherosclerosis (KUMA) study [33] demonstrated a significant association between a recent history of cancer and the risk of recurrent coronary atherosclerosis in populations undergoing PCI, and showed that a malignancy status can predict the likelihood of cardiovascular events following PCI, whereby malignancy was identified as an independent predictor of target lesion revascularisation (TLR) (HR 2.28; 95% CI 1.30–4.0, *p* = 0.004) [33]. Table 1 provides a summary of recent studies looking at outcomes following PCI amongst cancer patients.

## 4. Multidisciplinary Approach

A crucial factor to consider prior to offering patients PCI is whether they present with stable or unstable features. From randomised controlled trials, we know that PCI has not been shown to have a prognostic impact in the elective stable angina setting. For instance, COURAGE found that as an initial management strategy in patients with stable CAD, PCI did not reduce the risk of death (19.0% vs. 18.5%) or MI (13.2% vs. 12.3%) when added to optimal medical therapy at 4.6 years [41]. More recently, ISCHEMIA demonstrated that among patients with stable CAD and at least moderate ischemia, an initial invasive strategy did not reduce the risk of ischemic cardiovascular events or death from any cause over a period of 3.2 years (5-year cumulative event rate 16.4% vs. 18.2%) [42]. However, these trials did not specifically look at cancer patients and, in fact, ISCHEMIA excluded patients with a life expectancy shorter than the trial duration, which suggests that a proportion of cancer patients may have been excluded from the study. A reasonable pragmatic approach would be to aim to manage cancer patients who have stable angina using optimised medical therapy, especially if they are likely to require surgical resection of their cancer, or chemotherapy, which may be complicated by a dual antiplatelet regime.

In patients presenting with ACS, operators will have to carefully weigh the benefits against the risks of offering an invasive PCI procedure to these patients. In particular, issues to consider are the prognosis and estimated survival of patients, their particular risk of complications, such as thrombosis vs. bleeding, and any imminent plans for cancer-related interventions, such as surgical resection or biopsy, which would necessitate premature interruption to an antiplatelet regime. In order to be able to offer the best evidence-based, holistic, and tailored care to these patients, a multidisciplinary approach that brings together expertise from clinical/radiation oncologists, haematologists, cardiologists, surgeons, as well as the palliative team, is of prime importance.

## 5. Vascular Access

An important consideration when treating cancer patients with PCI is vascular access, which should be meticulously planned. In the absence of contraindications, such as anticipated haemodialysis, a radial approach should be favoured in order to reduce bleeding risks and promote early ambulation. Ultrasound guidance and the use of smaller hydrophilic sheaths as well as anticoagulation for radial access are encouraged in an effort to reduce bleeding risk [43]. Whenever radial access is not a feasible option, transfemoral access using a combination of ultrasound and fluoroscopic guidance, together with a micro-puncture technique, should be utilised.

## 6. Pharmacology

The use of aspirin has been shown to be effective and safe in patients with cancer and coronary artery disease. Previous studies [44] have found aspirin to be independently predictive of improved survival, whilst other studies comparing the outcomes of AMI in patients with and without cancer have reported similar improved results of medical therapy including aspirin [45]. However, the presence of pro-thrombotic states and haematological abnormalities, such as anaemia and thrombocytopenia, frequently leads to challenges in decision-making surrounding the use of antiplatelet and anticoagulant therapy for patients undergoing PCI. The consensus document by the Academic Research Consortium for High Bleeding Risk (ARC-HBR) [18] recognises active malignancy (defined as a diagnosis within the previous 12 months or ongoing active cancer treatment) as a major ARC-HBR criterion and further identified anaemia and thrombocytopenia as independent risk factors for HBR at the time of PCI [18].

The use of P2Y12 inhibitors in this population of patients can thus be challenging due to concerns regarding bleeding risks. Subgroup analysis from HORIZONS-AMI (Harmonising Outcomes with Revascularisation and Stents in Acute Myocardial Infarction) reported that baseline in patients with thrombocytopenia with ST-elevation MI undergoing coronary angiography and PCI was strongly associated with early adverse events and was also a marker of late events related to both ischemia and bleeding. [46,47] In another study [48], chronic thrombocytopenia in patients undergoing PCI was reported to correlate with an increased risk of bleeding complications necessitating blood products, vascular complications, ischaemic stroke, and in-hospital mortality [47,48].

Furthermore, patients with active malignancy are frequently seen to be anaemic either as a direct result of their malignancy or secondary to cancer therapy. In contemporary cohorts of patients undergoing PCI, anaemia has been shown to be associated with significantly increased rates of bleeding, re-infarction, post-procedural death, as well as major adverse cardiac events (MACEs) [49]. Keeping this in mind and after carefully evaluating the bleeding risk using risk stratification scores such as PRECISE-DAPT [50], operators may have to select a duration shorter than 12 months of antiplatelets in patients undergoing PCI in non-elective settings.

The MASTER-DAPT study [51] sought to answer the questions surrounding the appropriate duration of DAPT therapy in patients at a high risk of bleeding following the implantation of drug-eluting stents. MASTER-DAPT included 935 (2%) patients with a previous or active history of cancer. The authors reported one month of dual antiplatelet therapy (DAPT) to be non-inferior to the continuation of therapy for at least 2 additional months with regards to the occurrence of adverse clinical events and major adverse cardiac or cerebral events, with shortened therapy also resulting in reduced incidence of major bleeding (BARC type 2 bleeding 4.5% in the abbreviated-therapy group vs. 6.8% in the standard-therapy group; 95% −3.59 to −0.90). [51] Thus, in cancer patients deemed at high risk for bleeding, an abbreviated DAPT strategy may provide a safe alternative to standard DAPT regimes, reducing the bleeding complications whilst maintaining the ischemic benefits. It is worth noting that the benefit of a shortened duration of DAPT in HBR patients appears to extend to complex PCI procedures, as reported in a recent MASTER DAPT sub-analysis [52]. The investigators assessed the effects of 1- or ≥3-month DAPT in HBR patients receiving sirolimus-eluting stents for complex PCI (defined as one of the following criteria: three vessels treated, ≥3 stents implanted, ≥3 lesions treated, bifurcation or chronic total occlusion lesions, or total stent length ≥ 60 mm). It was reported that an abbreviated DAPT regime was not associated with a significantly higher risk of composite or individual ischemic events compared to standard DAPT among HBR patients undergoing complex PCI, with the additional benefit that an abbreviated DAPT regime resulted in significantly lower major bleeding complications compared to standard regimes (HR:0.64; 95%:0.42–0.9) [52].

Two important trials compared 1 month of DAPT therapy with at least 12 months of DAPT therapy after PCI using drug-eluting stents. The GLOBAL LEADERS trial [53] demonstrated that 1 month of a DAPT regime followed by ticagrelor monotherapy for an additional 23 months was non-inferior but not superior to the standard 12 months of DAPT therapy followed by aspirin monotherapy for 12 months, with the composite outcomes and major bleeding being similar between the two groups. The post-hoc analysis within 12 months showed a reduction in the composite of all-cause death and new Q wave MI in the experimental cohort, suggesting a potential advantage of stopping aspirin at 1 month followed by ticagrelor monotherapy. The TWILIGHT trial [54] examined the effect of ticagrelor alone as compared to ticagrelor plus aspirin with regard to clinically important bleeding amongst patients who are at high risk for bleeding or ischemic events post-PCI. TWILIGHT found that amongst high-risk patients undergoing PCI having completed 3 months of DAPT, ticagrelor monotherapy was associated with a lower incidence of relevant bleeding compared to the combination of ticagrelor and aspirin, with no heightened risks of death, myocardial infarction, or stroke [54]. This study thus suggests that ticagrelor monotherapy following an abbreviated course of DAPT may be safer in terms of bleeding risks amongst cancer patients undergoing PCI, whilst simultaneously preserving the ischemic benefits in that class of HBR patients.

In the STOPDAPT-2 trial, 1 month of DAPT therapy followed by clopidogrel monotherapy, compared to the standard 12 months of DAPT with aspirin and clopidogrel led to significantly lower rates of cardiovascular/bleeding event composites, which met the criteria for both non-inferiority and superiority. These findings suggest that a shortened DAPT duration may prove beneficial in HBR patients, such as individuals with active cancer. The LEADERS FREE trial [55] compared biolimus-coated stents with bare-metal stents (BMS) in patients at a high risk of bleeding and found that the drug-coated stent was superior to the BMS with respect to composites of cardiac death, MI, or stent thrombosis and target-lesion revascularisation when used with a 1-month course of DAPT [55]. It is notable that the rates of myocardial infarction were significantly lower in the drug-coated stent cohort. In the Onyx-One trial, HBR patients were randomised to Zotarolimus-eluting stents (ZES) vs. Bio Freedom polymer-free biolimus stents (DCS). The study reported that amongst patients with HBRs treated with 1-month DAPT followed by single antiplatelet therapy, the Onyx ZES had similar 2-year outcomes for the composite of cardiac death, myocardial infarction, or stent thrombosis at 1 year compared to the DCS. Table 2 provides a contemporary comparison of recent studies examining a 1-month duration of DAPT following PCI [52,53,55,56,57,58,59].

Based on the current evidence, the European guidelines recommend shortening DAPT therapy in high-bleeding-risk patients, and this is irrespective of their clinical presentation (i.e., stable angina or acute coronary syndrome) to six months or less. In certain cases, such as patients with stable coronary artery disease in whom 3-month DAPT poses safety concerns, DAPT for 1 month may be considered (level IIb evidence) [60]. Furthermore, the International DCB Consensus Group [61] highlighted the advantages conferred by DCB over stent implantation in HBR groups, with the recommended duration of DAPT being 1 month following a DCB-only strategy in de-novo vessels, with good results in patients with stable conditions [62,63]. Finally, proton-pump inhibitors (PPI) are recommended in an effort to minimise the risk of serious gastrointestinal (GI) haemorrhage, which is the most serious bleeding complication resulting from the use of long-term antiplatelet therapy [60].

## 7. Thrombocytopenia

The SCAI expert consensus document [64] makes a number of recommendations for cancer patients with thrombocytopenia undergoing coronary interventions. Prophylactic platelet transfusion is not routinely encouraged, unless specifically advised by the oncology/haematology team, in scenarios which may include a platelet count < 20,000/mL accompanied by pyrexia, leucocytosis, a rapidly declining platelet count, or other clotting deficiencies [64]. In patients with a platelet count of < 50,000/mL, unfractionated heparin at a dose of 30–50 U/kg is recommended, with regular activated clotting time (ACT) monitoring. Aspirin may be used with a platelet count of < 10,000/mL, whilst DAPT therapy with clopidogrel may be used with platelet counts between 30,000 and 50,000/mL. The use of other P2Y12 inhibitors, such as ticagrelor and prasugrel, and the use of IIb-IIIa inhibitors are not currently recommended in patients with a platelet count of < 50,000/mL [64].

## 8. Intravascular Physiological Assessment

The use of intracoronary physiological indices, such as Fractional Flow Reserve (FFR), which is the ratio of [65] mean distal coronary pressure to mean aortic pressure, remains an important means to guide revascularisation strategy in intermediate lesions and multivessel coronary disease and has a key role in assessing whether patients would benefit from revascularisation vs. optimised medical therapy [66]. Thus, in high-risk cancer patients with angiographic disease determined to be physiologically non-significant, this can avoid unnecessary stent implantation and eliminate the thrombotic and bleeding risks associated with this particular group of patients. An FFR of ≤ 0.80 generally implies a haemodynamically substantial stenosis. A study by Donisan et al. [67] suggested that amongst cancer patients, the treatment of haemodynamically significant obstructive lesions based on an FFR of ≤ 0.75 was correlated with a significant mortality benefit compared to deferring revascularisation [67].

## 9. Stent Choice

For cancer patients who are deemed high bleeding and for whom a shortened course of DAPT therapy is likely to be offered, a meticulous stent strategy has to be planned. Historically this has involved a relatively conservative approach with balloon angioplasty (POBA) or bare-metal stents (BMS). [43] According to the Society for Cardiovascular Angiography and Intervention (SCAI), POBA alone can still be considered for platelet counts of < 30,000/mL [68].

In recent times, a couple of randomised controlled trials have demonstrated that new-generation drug-eluting stents (DES) [58,69] are superior to BMS in populations with HBR where long-term DAPT therapy is not an option. Novel DES platforms, including the polymer-free and carrier-free BioFreedom stent, have been reported to be superior to bare metal stents (BMS), with a shortened duration of (DAPT) of 1 month [55]. The ONYX-one study by Windecker et al. [56] demonstrated that amongst patients at an enhanced risk of bleeding, 1 month of DAPT therapy following PCI with zotarolimus-eluting stents was comparable to the use of polymer-free drug-coated stents when looking at safety as well as adverse outcomes [25,56]. Furthermore, the TWILIGHT study [54] showed that amongst high-risk patients treated with PCI and having completed 3 months of DAPT therapy, ticagrelor monotherapy correlated with a lower incidence of significant bleeding when compared to ticagrelor plus aspirin, with no increased risk of death, stroke, or myocardial infarction [54]. In addition, a study looking at roughly 1600 patients with stable or unstable angina, showed that DES implantation using second-generation Zotarolimus-eluting stents, combined with a shortened, tailored DAPT regime, was found to carry a lower risk of MACE at 12 months amongst those patients considered uncertain candidates for DES. [69] Thus, in patients with a platelet count of > 30,000/mL and for whom there is no imminent need for surgery, a newer generation DES is recommended [68].

In cancer patients undergoing thoracic external beam radiation therapy (EBRT), one study showed that this was not associated with increased stent failure rates when used before or after PCI. [70] Thus, having stents does not preclude cancer therapy with EBRT in this group pf patients. However, higher rates of in-stent restenosis were reported amongst patients undergoing BMS as compared to DES. [70] Where safe and feasible, a relatively conservative approach to PCI, whereby bifurcation stenting and overlapping stents are avoided, may help reduce the risk of future stent thrombosis [64].

## 10. Intracoronary Imaging

In patients with cancer, the use of intravascular imaging adjuncts can help recognise those patients with acceptable minimum lumen areas (MLAs) for whom it may be safe to postpone revascularisation [43]. This has been examined in previous studies, including that by de la Torre Hernandez et al., where an MLA of 6 mm or more was reported to be a safe value for deferring the revascularisation of left main lesions [71]. This can be helpful in patients with left main stem disease but was deemed high risk for percutaneous intervention due to frailty, cancer burden, or associated bleeding risks. However, remaining vessel coronary lesions should be assessed using physiological indices such as fractional flow reserve (FFR) or instantaneous wave-free ratio (iFR).

In patients undergoing PCI, intravascular imaging in the form of intravascular ultrasound (IVUS) or optical coherence tomography (OCT) can define vessel architecture [72] and has an emerging key role in detecting and quantifying coronary atheroma, thrombus and calcium burden, [73] assessing stent expansion and malapposition, and excluding periprocedural complications, such as edge dissection. Two recent exert consensus position statements from the European Association of Percutaneous Coronary Interventions (EAPCI) highlight the role of imaging in guiding and optimising stent implantation [72,73,74]. OCT, which has superior special resolution as compared to IVUS (but lesser depth of penetration), can delineate thrombus, plaque rupture, stent under-expansion, edge dissections, and stent margin disease, which can guide the operator in terms of placing more stents if required or further post-dilating existing stents [75]. In the PROTECT-OCT Registry [76], the use of OCT in cancer patients undergoing PCI enabled operators to highlight high-risk patients based on whether they had uncovered stent struts, stent underexpansion, malapposition, or in-stent restenosis. Thus, OCT imaging enabled the identification of low-risk cancer patients with DES implanted who may safely prematurely cease DAPT in order to have cancer-related surgery or interventions [76].

## 11. Conclusions

As therapies for cancer continue to improve and patients live longer, the prevalence of comorbid chronic conditions, such as coronary artery disease, continues to be on the rise. The intersection of cancer and coronary artery disease can prove very challenging for clinicians, especially when considering percutaneous interventions. In particular, older age, frailty, comorbid burden and haematological abnormalities represent some of the factors that need meticulous consideration and a balanced approach to PCI in this category of patients. At the core of management should be a continuous risk–benefit assessment, clear patient/multidisciplinary discussion, and the utilisation of all the latest evidence and technology available in order to provide the best personalised treatment to cancer patients.

## Figures and Tables

**Table 1 jpm-12-01372-t001:** Summary of recent studies looking at outcomes following PCI amongst cancer patients.

Study (Year)	Population	Outcomes following PCI
Thomason et al. (2022) [34]	Case-control study of 2016 US NIS using machine learning to examine 30,195,722 hospitalized patients, of whom 1.27% had gynaecological cancer of whom 0.02% underwent PCI.	Among gynecological cancer patients, mortality was significantly reduced by undergoing PCI (OR 0.58, 95% CI 0.39–0.85; *p* = 0.006). PCI reduced mortality but not significantly for metastatic patients (OR 0.74, 95% CI 0.32–1.71; *p* = 0.481)
Mohamed et al. (2021) [35]	US NIS population hospitalised with STEMI from 2004 to 2015 with estimation of treatment effect of PPCI amongst cancer patients. (cancer N = 38,932)	Patients with cancer received PPCI less frequently (54–70% vs. 82% in no cancer). PPCI associated with lower adjusted probabilities of MACCE and all-cause mortality in the cancer groups compared with the no cancer group
Kwok et al. (2021) [25]	US Nationwide Readmission Database from 2010 to 2014 (cancer N = 51,289)	90-day readmission after AMI following PCI was higher in patients with active cancer (7–12% vs. 5.6% in no cancer) 90-day readmission for bleeding post PCI was higher in cancer patients (0.6–4.2% vs. 0.6% in patients with no cancer.
Potts et al. (2020) [29]	US NIS sample of leukemia patients undergoing PCI between 2004 and 2014, CLL accounting for 75% of the cohort. (cancer N = 15,789)	After multivariable adjustment, leukemia patients had significantly increased odds of in-hospital mortality (OR 1.41, 95% CI 1.11–1.79) and bleeding (OR 1.87, 95% CI 1.56–2.09).
Velders et al. (2020) [36]	National Swedish quality registries- all patients admitted with AMI between 2001 and 2014 (cancer N = 16,237).	During a median follow up of 4.3 years, cancer was associated with all-cause mortality (HR 1.44, 95% 1.40–1.47), recurrent MI (HR 1.08, 95% CI 1.04–1.12), heart failure (HR 1.10, 95% CI 1.06–1.13) and major bleeding (HR 1.45, 95% CI 1.34–1.57)
Potts et al. (2019) [24]	US NIS population undergoing PCI between 2004 and 2014, having current and previous cancer rates of 1.8% and 5.8% respectively.	Patients with current lung cancer had greater in-hospital mortality (OR 2.81, 95% CI 2.37–3.34) and any in-hospital complication (OR 1.21, 95% CI 1.10–1.36), while current colon cancer was associated with any complication and bleeding.
Borovac et al. (2019) [37]	US NIS sample patients undergoing PCI between 2004 and 2014, 0.25% of which had a diagnosis of lymphoma (N = 18,052).	Lymphoma was associated with increased odds of in-hospital mortality (OR 1.39, 95% CI 1.25–1.54), stroke or transient ischemic attack (OR 1.75, 95% CI 1.61–1.90) and any in-hospital complication (OR 1.31, 95% CI 1.25–1.27), following PCI.
Tabata et al. (2018) [33]	Kumamoto University malignancy and Atherosclerosis Study (KUMA) study looking at cancer and no-cancer patients undergoing PCI	The malignant group had had a significantly higher probability of TLR than the non-malignant group (*p* = 0.002), with proportional hazards analyses identifying malignancy as an independent predictor of TLR (HR 2.28, 95% CI 1.3–4.0; *p* = 0.004)
Landes et al. (2017) [38]	Retrospective analysis of 12785 patients undergoing PCI (1005 cancer patients), Rabin institution, Israel.	Cancer patients had increased all-cause mortality (HR 1.46, 95% CI 1.24–1.72), death or non-fatal MI (HR 1.40, 95% CI 1.20–1.64).
Guddati et al. (2016) [39]	US NIS looking at patients with metastatic cancer (N = 49,515) and ACS between 2000 and 2009.	The adjusted odds of receiving PCI in patients with metastatic disease increased by 1,14 every year in the last decade, with the beneficial effect of PCI on in-hospital mortality having been noted to decline in NSTEMI such that by 2009, there was no significant difference between patients who received PCI and those who did not.
Wang et al. (2016) [40]	Retrospective cohort in Mayo clinic PCI registry of 2346 patients admitted with STEMI between 200 and 2010 (cancer N = 261)	At 5 years, patients with cancer had similar cardiac mortality (4.2% vs. 5.8%; HR=1.27; 95% CI, 0.77–2.10; *p* = 0.35), despite being seen to experience more heart failure hospitalizations (15% vs. 10%; HR=1.72; 95% CI, 1.18–2.50; *p* = 0.01). Cancer was associated with higher in-hospital non-cardiac mortality (1.9% vs. 0.4%, *p* = 0.03)
Hess et al. (2015) [11]	Stented patients at Duke (1996–2010) using the Duke Information Systems for Cardiovascular care and the Duke Tumor Registry.(total patients 15,008; cancer N = 496)	Post-PCI, cancer was associated with cardiovascular mortality (adjusted HR= 1.51; 95% CI 1.11 to 2.03). Composite cardiac death, MI or revascularisation was similar between pre-PCI cancer and no-cancer patient groups.

ACS—acute coronary syndrome; HR—hazards ratio; MI—myocardial infarction; OR—odds ratio; PCI—percutaneous coronary intervention; TLR—target lesion revascularisation.

**Table 2 jpm-12-01372-t002:** Studies assessing 1-month DAPT following PCI.

Study	Design	Population	Stent Type	Follow-up (Months)	Primary Outcome	Result
ONYX ONE (2020) [56]	Single blinded RCT	1996	Resolute Onyx vs. Biofreedom polymer-free DCS	12	Cardiac death/MI/ST	Primary outcome for Resolute Onyx vs. Biofreedom 17.1% vs. 16.9%, *p* = 0.011 for non-inferiority
STOP-DAPT-2 (2019) [57]	Open-label RCT	3045	Xience DES	12	Composite death, ST and bleeding in 1-month vs. 12-month DAPT	Composite end-point of 1-month vs. 12-month DAPT 2.4% vs. 3.7%, *p* < 0.001 for non-inferiority
GLOBAL-LEADERS (2018) [53]	Double-blinded RCT	15 968	Biolimus eluting stent	24	All-cause mortality+ MI	Primary outcome for 1-month DAPT followed by 23months ticagrelor vs. 12-month DAPT 3.8% vs. 4.4%, *p* = 0.073
LEADERS FREE (2015) [55]	Double-blinded RCT	2466	Biofreedom DCS vs. BMS	13	Composite cardiac death/MI/ST for Biofreedom vs. BMS	Primary outcome for Biofreedom vs. BMS 9.4% vs. 12.9%, *p* < 0.001 for non-inferiority
ZEUS (2015) [58]	Single-blinded RCT	1606	ZES vs. BMS	12	MACE at 12months (all-cause mortality, MI, TVR)	ZES vs. BMS 17.5% vs. 22.1%, *p* = 0.011
MASTER DAPT [52]	RCT	4579 HBR patients	Bioresorbable polymer-sirolimus eluting stent	12	Composite all-cause death, MI, stroke or major bleeding	Net adverse clinical events 7.5% in abbreviated vs. 7.7% in standard group
XIENCE 90/28 [59]	RCT	3652 HBR patients	1-month (XIENCE 28 USA/Global) or 3-months (XIENCE 90) vs. standard DAPT for 12months	12	Composite all-cause death/MI	1-month DAPT associated with similar ischemic outcomes (7.3% vs. 7.5%) and lower BARC2-5 bleeding risks (7.6% vs. 10.0%)

BARC—Bleeding Academic Research Consortium; BMS—bare metal stent; DAPT—dual antiplatelet therapy; DCS—drug-coated stent; DES—drug-eluting stent; HBR—high bleeding risk; MI—myocardial infarction; PCI—percutaneous coronary intervention; RCT—randomised control rial; TLR—target lesion revascularisation; ZES—zotarolimus-eluting stent.

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
