# Peer review of "Percutaneous Coronary Angioplasty in Patients with Cancer: Clinical Challenges and Management Strategies"

_jpm, 2022, doi:10.3390/jpm12091372_

Round 1
Reviewer 1 Report
The article reviewed current status of PCI in cancer patients, which was an extremely important issue. The review was extensive and up to date.
Author Response
The article reviewed current status of PCI in cancer patients, which was an extremely important issue. The review was extensive and up to date.
We thank the Reviewer for their acknowledgement of the importance of this topic, as well as their appreciation of this review.

Reviewer 2 Report
I think this work is very important in the field. The authors can follow some comments below to improve their mansucript.
1. The title should not be abbreviated.
2. Please included some figures in this work.
3. All abbreviations should be defined in a separate table.
4. Comparison tables should be added.
Thank you!
Author Response
I think this work is very important in the field. The authors can follow some comments below to improve their manuscript.
We thank the reviewer for their comment and appreciation of the importance of this topic
- The title should not be abbreviated.
We have modified the title to remove abbreviations – Please see Page 1 of the main manuscript.
- Please included some figures in this work.
We thank the Reviewer for their suggestion. Please see the illustration included with this submission.
- All abbreviations should be defined in a separate table.
We have now included a separate table of abbreviations- Please see Page 2 of the main manuscript.
- Comparison tables should be added.
We thank the Reviewer for their suggestion. Please find attached document with 2 new tables- the first summarising clinical outcomes following PCI amongst patients undergoing PCI (Table 1- referred to on page 8) as well as a second table summarising the recent studies assessing outcomes of 1-month dual antiplatelet therapy (table 2-referred to on page 12)duration among patients at high bleeding risk (such as cancer patients) vs standard therapy following percutaneous coronary angioplasty.
